# The impact of Public Health Emergency (PHE) on the news dissemination strength: Evidence from Chinese-Speaking Vloggers on YouTube

**Dan Sun** [1], **Guochang Zhao** [2]*

**1** School of Economics, Hunan Institute of Engineering, Xiangtan, Hunan Province, China, **2** Research Institute of Economics and Management, Southwestern University of Finance and Economics, Chengdu, Sichuan Province, China

* guochangzhao@swufe.edu.cn

## Abstract

News dissemination plays a vital role in supporting people to incorporate beneficial actions during public health emergencies, thereby significantly reducing the adverse influences of events. Based on big data from YouTube, this research study takes the declaration of COVID-19 National Public Health Emergency (PHE) as the event impact and employs a DiD model to investigate the effect of PHE on the news dissemination strength of relevant videos. The study findings indicate that the views, comments, and likes on relevant videos significantly increased during the COVID-19 public health emergency. Moreover, the public's response to PHE has been rapid, with the highest growth in comments and views on videos observed within the first week of the public health emergency, followed by a gradual decline and returning to normal levels within four weeks. In addition, during the COVID-19 public health emergency, in the context of different types of media, lifestyle bloggers, local media, and institutional media demonstrated higher growth in the news dissemination strength of relevant videos as compared to news & political bloggers, foreign media, and personal media, respectively. Further, the audience attracted by related news tends to display a certain level of stickiness, therefore this audience may subscribe to these channels during public health emergencies, which confirms the incentive mechanisms of social media platforms to foster relevant news dissemination during public health emergencies. The proposed findings provide essential insights into effective news dissemination in potential future public health events.

## Introduction

Recently, there have been frequent public health emergencies, causing substantial adverse influences on public health. In particular, the outbreak of COVID-19 in 2019 triggered a significant number of infections and deaths on a global level. By the end of July 2023, the World Health Organization (WHO) reports that there have been over 768 million confirmed cases of COVID-19 across the world, with 6.95 million reported deaths [1]. In addition to the direct

**Funding:** The authors received no specific funding for this work.

**Competing interests:** The authors have declared that no competing interests exist.

effect of the virus on public health, the preventive measures incorporated to control the pandemic, such as lockdown and social distancing, have disrupted the normal social gatherings and interactions, leading to higher levels of sleep disorders, anxiety, stress, and depression, consequently contributing to a rise in mental health concerns [2–4]. Moreover, the COVID-19 pandemic has intensively strained the healthcare systems around the globe, thereby stimulating not only resource shortages but also negatively affecting the medical services for non-COVID-19 patients [5,6]. At the same time, certain patients may have postponed seeking medical care during the pandemic due to concerns about virus infection, thus resulting in the neglect and deterioration of other health matters [7]. Owing to the severe negative impacts of public health emergencies on public health, the public actively seeks information about these events through various channels with the expectation of quickly understanding the situation to protect their own health, as well as that of their family and friends [8,9]. Therefore, the effective dissemination of information to the public through media during the early stages of a public health emergency emerges as a topic of shared concern for both policymakers and research scholars.

News media play a vital role in disseminating public health and policy information during public health emergencies [10–14]. This allows the relevant authorities to instantly share the latest updates, guidelines, and safety precautions with the general public, consequently aiding in controlling diseases and informing the public regarding the latest developments [15]. In fact, news media not only convey potential risks but also influence public perceptions through the presented content, volume, and tone of media coverage [16,17]. Furthermore, through social media channels, the news sources not only effectively disseminate information but also promptly receive feedback from audiences, extending a framework for prevalent public debates concerning policy responses, including conflicting priorities related to the timing or stringency of policy implementation [11,12]. Besides this, news reports swiftly disseminate treatment protocols during public health emergencies at regional-, national-, and international levels [18]. This supports less-resourced facilities to quickly adopt protocols and incorporate essential adjustments to fit their distinct circumstances and accessible resources. Furthermore, news coverage facilitates isolated individuals to comprehend external information. As a result, news coverage mitigates the sense of isolation and boredom connected with anxiety, depression, and long-term distress, thus making it an essential resource for isolation at home to reduce the psychological impact [19]. Evidently, the significant influence of news dissemination is undeniable during public health emergencies, as witnessed during the COVID-19 pandemic, where news media played an imperative role in broadcasting information, influencing billions worldwide [20].

With the development of Information and Communication Technology, significant changes have taken place in the way and nature of news dissemination. Many scholars have systematically studied the characteristics of news dissemination during public health emergencies. First, time factors are considered as important factors affecting the news dissemination during public health emergencies [21]. In different life cycles, there are great differences in the propagation speed and diffusion volumes of event-related information on social media [22,23]. Second, media factors are also significant factors influencing the news dissemination during public health emergencies. The scope and diffusion degree of news on social media vary significantly based on the media type and its influence [24,25]. Additionally, the contents of news are also important influence factors for news dissemination in public health emergencies. For instance, when public health emergencies occur, there are often great differences in the dissemination of news under different topic categories on social media [26,27]. Similarly, for information with different sentiments and emotions, the dissemination of information presents different trends and states [28].

Though there is a consistent interest from different disciplines, studies on the general characteristics of news dissemination during public health emergencies demonstrate certain limitations. First, most existing studies have explored the general characteristics of news dissemination in public health emergencies through descriptive statistics, while causal analysis is relatively lacking [21]. Second, most research studies on news dissemination have primarily concentrated on English-speaking audiences or China's internet users, with relatively limited literature on overseas Chinese-speaking audiences and communities. Third, research data adopted in pertinent empirical studies frequently confront challenges associated with accuracy and efficacy. For instance, non-random sampling methods and small sample sizes can lead to survivor bias, selection bias, or other methodological drawbacks.

In order to bridge the research gap discussed above, this study aims to explore the impact of Public Health Emergency (PHE) on the news dissemination strength. Specifically, to achieve this goal, this research study adopted big data from Chinese-speaking channels on YouTube while employing the difference-in-differences (DiD) method to analyze the dynamic changes in the news dissemination strength of pertinent content during public health emergencies. In addition, this paper explored the heterogeneity in changes in news dissemination strength across different types of video bloggers. YouTube serves as the largest video-sharing website on the internet by hosting a plethora of video bloggers who share different types of videos. This platform also attracts a massive audience that watches video content and extends personal comments. As of Jan 2022, YouTube had reached 2.562 billion monthly active users across the world [29]. In Taiwan, Hong Kong, and countries/regions where overseas Chinese populations exceed 1 million, the proportion of YouTube users among the population aged 18 and above is remarkably high [30]. Therefore, the researchers believe that Chinese-speaking channels on YouTube serve as a widely engaged online social platform for overseas Chinese, Hong Kong, and Taiwan residents, with its micro-level data carrying strong representativeness. Thus, research undertaken on this platform can offer valuable insights into the common/general patterns of online news dissemination among overseas Chinese-speaking populations and communities.

This research article contributes to the extant literature in the following manners. Firstly, a comprehensive analysis of the changes in news dissemination strength of online media during public health emergencies is performed in this study using the difference-in-differences (DiD) approach and event study. In fact, the DiD approach and event study are commonly applied data analysis methods in the field of econometrics, which make the study results not only hold statistical significance but also extend explicit insights into the causal relationships between public health emergency and the variations in dissemination strength of relevant news. Secondly, this research analyzes the news dissemination of Chinese-speaking channels on YouTube during public health emergencies. The potential audience of these Chinese-speaking channels primarily includes Chinese-speaking internet users outside of mainland China. By examining the dissemination of relevant news among overseas Chinese-speaking communities during public health emergencies, this research paper offers a valuable supplement to the research on the general characteristics of news dissemination. Thirdly, this paper employs big data generated through random sampling in order to study the dynamic effects on the dissemination strength of relevant news content during public health emergencies. As a consequence, the abundant data resources enable the researchers to carry out multifaceted analyses. In specific, this study not only scrutinizes the impact of public health emergencies on the dissemination strength of relevant news but also ascertains the potential heterogeneity of the proposed impact among various types of YouTube channels, such as lifestyle bloggers, and news & political bloggers, institutional and personal media, and local and foreign media.

## Research background and relevant literature

### Research background

In Jan 2020, the COVID-19 pandemic rapidly spread and garnered a significant amount of global attention [31]. In this study, the development of China's COVID-19 outbreak is outlined based on the content of a white paper titled "Fighting COVID-19: China in Action," which was published by the China's State Council Information Office [32]. Primarily, the first phase of China's COVID-19 outbreak spanned from 27th Dec 2019, to 19th Jan 2020. On 27th Dec 2019, Hubei Provincial Hospital of Integrated Chinese & Western Medicine reported several cases of pneumonia from unknown causes to the Wuhan Jianghan Center for Disease Control and Prevention (CDC). Subsequently, on 31st Dec 2019, the Wuhan Municipal Health Commission released a situation report on its official website related to the existing pneumonia situation in the city, while confirming 27 active cases and instructing the public to wear masks in public places and avoid poorly ventilated, closed and crowded spaces. Thereafter, the Wuhan Municipal Health Commission renamed the "pneumonia of unknown cause" to "pneumonia caused by a novel coronavirus" in the situation report on 12th Jan 2020. Meanwhile, the National Health Commission (NHC) of China shared the genome sequence of the novel coronavirus (2019-nCoV) with the WHO, which was published by the Global Initiative on Sharing All Influenza Data to be shared across the globe.

Afterward, the 2nd stage of China's COVID-19 outbreak took place from 20th Jan 2020 to 20th Feb 2020. On 20th Jan, the NHC of China organized a press conference where a high-level team of medical experts confirmed that the said virus could be transmitted from one person to other individuals. Afterward, the NHC issued an announcement classifying COVID-19 as a Class B infectious disease under the Law of the People's Republic of China on Prevention and Treatment of Infectious Diseases and implementing control measures as if COVID-19 were a Class A contagious disease. The State Council convened a teleconference on the same day, in order to plan for nationwide control and prevention of the pandemic. On 23rd Jan, Wuhan declared temporary closure of the city's outbound routes at all airports and railway stations. From 23rd to 29th Jan, all provinces across the country activated Level 1 public health emergency response. Subsequently, the WHO declared the COVID-19 outbreak a public health emergency of international concern on 31st Jan. Thereafter, the number of cured and discharged cases in Wuhan outnumbered newly confirmed cases for the first time on 19th February.

The 3rd and 4th stages of China's COVID-19 outbreak were from 21st Feb to 28th April, during which the rapid spread of the virus had been contained in Wuhan and the rest of Hubei Province. Notably, the daily figure for new cases had remained in single digits since mid-March. Afterward, the restrictions on outbound traffic from Wuhan City and Hubei Province were lifted on 8th April, and the last hospitalized COVID-19 patient in Wuhan was discharged on 26th April. This shows that China initially halted the spread of COVID-19 on the mainland. The 5th stage of China's COVID-19 outbreak began on 29th April; thereafter, marking the transition to the normalization of pandemic prevention and control measures throughout China.

Prominently, although the WHO officially declared COVID-19 a public health emergency of international concern on 31st Jan, China's government effectively recognized the COVID-19 outbreak as a public health emergency, supported by the activation of nationwide public health emergency responses initiated starting from 20th Jan. Therefore, this research study assumes 20th Jan as the initiation of the COVID-19 public health emergency, while using the proposed event as a boundary to distinguish between the early phase of the COVID-19 outbreak and the phase of the public health emergency.

## Relevant literature

During the outbreak of COVID-19, social media became an important channel for the government to disseminate information to the public [33]. Most relevant studies primarily explored the characteristics of news dissemination during public health emergencies from the perspectives of time factors and media factors.

**Time factors affecting news dissemination during public health emergencies.** Public health emergencies do not emerge abruptly; they progress through a series of different stages and form different life cycles [34]. Similarly, the dissemination dynamics of event-related news on social media exhibit variations across different life cycle stages [21,35]. Concerning the correlation between news dissemination strength and the life cycle stages of public health emergencies, different studies have yielded diverse conclusions. Research specifically related to the COVID-19 event found that the dissemination strength of news on social media aligns with the trend of the COVID-19 outbreak in the real world [33]. However, studies with a differing view argue that with the rapid development and extensive application of the internet, news related to emergency events spread faster than ever before [36]. As a result, the news dissemination of public health emergencies generally demonstrate a shorter life cycle, and a briefer active period compared to the trend of real-life crisis events [37].

Moreover, there also exists a lack of sustained news coverage during public health emergencies. Regardless of the prevalent threats to public health, news coverage promptly dwindled succeeding the initial declaration of a public health emergency [16,38]. In fact, similar challenges in terms of media coverage were observed during the 2014 Ebola epidemic and the 2003 Severe Acute Respiratory Syndrome (SARS) outbreak [39]. This implies that despite the persistence of related risks and challenges, the public may misconstrue the urgency and significance of the matter, owing to the lack of persistent media coverage. The stated drawbacks remind the researchers that there is a need to further comprehend and explore the role of media during public health emergencies as these measures shall prove to be essential tools for combating future possible outbreaks.

To further understand the time factors affecting news dissemination during public health emergencies, this research examines the dynamic trends of news dissemination strength during the COVID-19 outbreak. This not only supplements to the previous research of the general characteristics of news dissemination during public health emergencies, but also further explores the reasons for the lack of persistence in news reporting during such events.

**Media factors affecting news dissemination during public health emergencies.** During public health emergencies, information published by different media types varies greatly in the scope of influence and degree of diffusion on social media [24]. On the one hand, the influence of media themselves is an important factor affecting news dissemination. The number of subscribers a media has serves as the most intuitive standard for measuring their influence, and media outlets with a large number of subscribers tend to have a wider reach and diffusion for their published news [24,25,40]. For instance, during public health emergencies, news dissemination strength varies among news media and celebrities, media organizations and ordinary individuals, celebrities and ordinary individuals [41,42]. On the other hand, the media's credibility and communication abilities significantly influence news dissemination during health emergencies. For example, a Twitter study suggested that during COVID-19, news from "Mainstream or Local News" sources had greater dissemination strength than "Government or Public Health" sources [43]. Therefore, government media can enhance their news dissemination effectiveness for public health emergencies by strengthening their credibility. Similarly, public health organizations need to improve their information dissemination capacity through better alignment with the general public's information needs during public health crises [44,45].

Although the influence of media types on news dissemination has attracted the attention of scholars, most of the previous studies were limited by insufficient data sources and were unable to make a more comprehensive analysis of media with different attribute characteristics. To address the aforementioned research gap, in the heterogeneity analysis section of this research article, media types are categorized as "news and political bloggers" and "lifestyle bloggers" from the perspective of content. Similarly, based on attribute characteristics of media, media types are classified as "local media" and "foreign media," "individual media" and "institutional media." This study subsequently examines the differences in news dissemination by different types of media during public health emergencies. This work not only complements previous research but also further unveils the general characteristics of news dissemination during public health emergencies.

## Data processing and research methodology

### Data collection

The data collection took place between 2nd June 2022 and 26th Aug 2022. During this period, we collected data through the YouTube API, which is publicly available, and for the analysis we used only public available data. The video channels/vloggers from which we collect data are public YouTube entities. User content from such entities is also publicly accessible unless restricted by their privacy settings, in which case it is not included in the dataset. Our collection and analysis method complied with the terms, conditions, and policies of YouTube. In order to obtain a research sample of video bloggers that fulfills the criteria of random sampling, we first randomly selected public video channels/bloggers, and then gathered all the video information which was accessible to the public within the selected channels. In total, this study collected 199,885 YouTube videos for a time period from September 2019 to June 2020. The dataset encompasses detailed information about video channels/vloggers, including their number of subscribers, language preference (Simplified or Traditional Chinese), and duration of media operation among others. Additionally, the dataset also offers metrics for individual videos, detailing titles, view counts, like counts, comment count, and release dates. During the process of data collection and processing, the following principles were observed:

(1) The approach for building the sampling frame of YouTube video bloggers has been established by repetitively searching for top keywords from top news in mainland China; and thereafter, obtaining video blogger information from the search results. Firstly, the top keywords were extracted from main events in mainland China for the time period from 2019 to 2020 [46,47]. Correspondingly, examples of these keywords can be found in S1 Table. Secondly, video information corresponding to each keyword was retrieved on a daily basis from September 2019 to June 2020 through the application programming interface offered by YouTube API Services. And video blogger information was obtained from these results. Since the YouTube API Services returns a maximum of 50 video information results for each keyword search, therefore the search for a single keyword on a specific day was repeated until no new video information was returned by YouTube API Services. By utilizing the aforementioned sampling method, the study sample includes the vast majority of Chinese-speaking YouTube video bloggers who have posted videos associated with mainland China between 2019 and 2020, even when not all Chinese-speaking bloggers are included in the sample. This ensures that the study sample is as highly close to the entire population or at least meets the criteria of random sampling for Chinese-speaking video bloggers on YouTube.

(2) Video bloggers were selected from all Chinese-speaking channels on YouTube with a higher level of relevance to mainland China. Specifically, through text analysis of video titles, descriptions, and tags, video bloggers who had 10% or more of their videos related to mainland

China topics were selected. Owing to the central focus of this paper on the COVID-19 Public Health Emergency (PHE) within mainland China, the sample scope of this study is limited to vloggers with a more pronounced linkage to topics concerning mainland China. This strategy extends inherent benefits by ensuring the comprehensive inclusion of primary disseminators of related news within the sample, while simultaneously avoiding undue expansion of the research sample. In the robustness analysis section, various samples were provided which were generated based on varying criteria for judgment.

(3) The video bloggers were selected from the categories of "People & Blogs," "News & Politics," and "Travel & Events" on YouTube. In fact, YouTube categorizes video content into 15 major classifications, including "News & Politics," "Film & Animation," and "People & Blogs," among others. Among these categories, "News & Politics" bloggers play a vital role in spreading COVID-19 pandemic-related information, followed by "People & Blogs" and "Travel & Events" bloggers. This study directly illuminates the dynamics of news dissemination relevant to the COVID-19 Public Health Emergency (PHE) by studying the videos published by these three categories of bloggers.

(4) The researchers selected channels that remained active from 31st Dec 2019 to 20th Feb 2020, as a sampling base. The focus of this research is on the COVID-19 PHE. The earliest videos on the COVID-19 pandemic from Chinese-speaking YouTube channels trace back to 31st Dec 2019. Similarly, the COVID-19 outbreak was officially declared a national Public Health Emergency (PHE) on 20th Jan 2020. Further, China's public health emergency was gradually being controlled, as of 20th Feb 2020, therefore, the period from 31st Dec 2019 to 20th Feb 2020 is chosen as the timeframe of interest for this research study.

(5) The samples with fewer than 10,000 subscribers are excluded from the sampling base. Primarily, this exclusion serves to resolve the concern of non-random sampling resulting from selective content distribution on YouTube. From the viewpoint of YouTube's content distribution mechanism, there exists a strong selective content distribution when promoting videos from smaller channels. Moreover, in the section on robustness analysis, regression results are also provided based on the samples with subscribers greater than or equal to 50,000 and 1,000 subscribers, respectively.

(6) From the aforementioned channels, videos linked to mainland China topics published between 31st Dec 2019, and 20th Feb 2020 are taken as video sampling base. Since the COVID-19 epidemic initially surfaced in mainland China, therefore the relevant news at first focused on the development of the pandemic situation in mainland China. Furthermore, the sample scope has been limited to videos concerning mainland China topics, in order to satisfy the parallel trends assumption. Eventually, the final sample consisted of 17,018 entries of YouTube video information. In the subsequent sections, this study presents empirical results for sample periods ranging from 1st Sept 2019 to 28th April 2020. Consistent with this, in the robustness analysis section, samples of videos unrelated to mainland China topics are included, demonstrating results similar to the baseline regression estimations.

## Data processing

This study first categorizes video bloggers as either personal media or institutional media based on their ownership structure. In accordance with their geographical location, video bloggers are also classified as local media (mainland China's media) or foreign media. At the same time, these bloggers are categorized as news & politics bloggers or lifestyle bloggers (including "Travel & Events" and "People & Blogs" bloggers), consistent with their primary video content. In this paper, personal media refers to entities primarily operated by individuals, with independent personal channels for interactive communication, whereas institutional

media consists of media companies, official media, and other media groups [48]. In the study sample, personal media includes bloggers who record their feelings, inscribe travelogues, and share food experiences, "grassroots reporters" who individually gather and comment on news, among others. Similarly, institutional media encompasses channels managed by local and national media entities on YouTube and the programs operated by other media organizations and groups.

The second step comprises data processing of the video content. In this study, video content is identified based on video titles, descriptions, and tags. For this purpose, the "Jieba" Chinese text segmentation module is employed to perform word segmentation on the required text content of the videos. Subsequently, this segmented text is matched with the predefined "Mainland China Corpus" and "COVID-19 PHE Corpus", in order to identify videos related to mainland China and the COVID-19 Public Health Emergency (PHE). For instance, the title is decomposed by the accurate mode of the "jieba" word segmentation into words such as "China," "succeeded in," "CDC," "isolating," "the," "first," "novel coronavirus strain," etc., when a video title is "China CDC succeeded in isolating the first novel coronavirus strain". Afterward, this title is matched with the vocabulary from the two aforementioned corpora. As a result, words such as "China" and "novel coronavirus strain" are successfully matched with existing terms in the "Mainland China Corpus" and "COVID-19 Epidemic Corpus" respectively. Consequently, such a video is characterized as both relevant to mainland China topics and linked to the COVID-19 pandemic subject. Owing to the lag and limitations of the Jieba segmentation library, which may pose challenges in handling specific "new" vocabulary, this research paper optimized the Jieba segmentation corpus using other vocabulary sources such as "China Scenic Spots Thesaurus" and "Chinese Thesaurus of Geographic Names" extended by Sogou, thus enhancing Jieba's segmentation accuracy. Through testing within the context of the study samples, this text processing technique yielded an accuracy of 93.2% in identifying topics associated with mainland China and an accuracy of 97.4% in selecting topics relevant to the COVID-19 pandemic.

## Research methodology

There has been a significant upsurge in the public's demand for information during the public health emergency, thereby leading to an evident increase in the dissemination strength of pandemic-relevant news. This change in news dissemination strength is evident in increased television viewership rates and, on video-sharing platforms like YouTube, translates to a significant rise in views, likes, and comments on relevant videos. In order to address possible endogeneity problems and eliminate interference from other factors, this research utilizes big data from YouTube and employs the DiD approach to investigate the trends of changes in news dissemination strength associated with COVID-19 prior to and during the public health emergency.

Particularly, the declarations of the national public health emergency in China are considered in this paper as a policy shock. Based on the difference-in-differences (DiD) approach, this research article estimates the growth in the number of views, likes, and comments on relevant videos stimulated by this policy shock, in order to explore the influence of the COVID-19 public health emergency on the dissemination strength of pandemic-related news. Correspondingly, Eq (1) represents the concrete regression estimation:

$$Y_{ijtl} = \beta_0 + \beta_1 \cdot post_i + \beta_2 \cdot \text{COVID} - 19\,\text{News}_{ijtl} + \beta_3 \cdot (\text{COVID} - 19\,\text{News}_{ijtl} \cdot post_i) + \beta_4$$
$$\cdot X'_{ijtl} + u_j + u_t + u_l + \varepsilon_{ijdl}(1)$$

Where $Y_{ijtl}$ stands for the measure of news dissemination strength, which in this paper is represented by the logarithm of the number of views, likes, and comments for each video; subscripts i, j, t, and l signify videos, YouTube channels, date, and topics, respectively. Additionally, $post_i$ serves as a dummy variable that takes the value 1 when video i is posted during the public health emergency (starting from 20[th] Jan 2020), and 0 otherwise. Further, COVID-19 $News_{ijtl}$ denotes a treatment group dummy variable that takes the value 1 when the content of video i is related to the public health emergency, and 0 otherwise. In this study, COVID-19 $News_{ijtl}$ equals 1 specifically refers to news coverage associated with the COVID-19 outbreak in mainland China. Similar videos in terms of content to the treatment group are chosen as the control group, in order to ensure the parallel trends assumption, where COVID-19 $News_{ijtl}$ equals 0 is applied to all other video content relevant to mainland China topics. Moreover, $X'_{ijtl}$ represents control variables, which contain information such as the number of subscribers for the video bloggers, video channels/blogger location, language (Simplified Chinese/Traditional Chinese), and years of media operation. Furthermore, $u_j$, $u_d$, and $u_l$ show the fixed effects for YouTube channels, publication dates, and video topics, respectively. Since the fixed effects for video publication dates and topics are controlled, the dummy variables $post_i$ and COVID-19 $News_{ijtl}$ are absorbed, and their coefficients shall not appear in the regression results.

In this study, $post_i$ constitutes the first difference in the DiD framework. On the one hand, the COVID-19 outbreak has not yet been recognized as a public health emergency, with no implementation of nationwide prevention and control measures when $post_i$ equals 0. In the same vein, public attention to the COVID-19 outbreak remains at a regular level. In contrast, the COVID-19 outbreak has been officially declared as a national public health emergency when $post_i$ equates to 1, thus, leading to prompt nationwide emergency response, including lockdowns of cities with populations exceeding 10 million, such as Wuhan. In line with this, public attention to this event is expected to rise substantially, with news coverage related to this event attaining a higher level of dissemination strength.

The second difference in the DiD framework pertains to the discrepancy in the dissemination strength of news with different themes during the COVID-19 public health emergency. Notably, video content related to COVID-19 is expected to receive more attention during public health emergencies, while the dissemination strength of video content on other topics shall not prominently change. In other words, there should be a relatively greater increase in the dissemination strength of videos during the public health emergency for samples where COVID-19 $News_{ijtl}$ is set to 1, indicating news associated with the COVID-19 outbreak. Conversely, there is no significant difference in news dissemination strength before or during public health emergency for samples where COVID-19 $News_{ijtl}$ is equal to 0, representing video content of no-COVID-19 themes. This constitutes the second difference in the difference-in-difference (DiD) model. Specifically, the DiD approach attempts to assess whether there exists a significant variance in the dependent/explained variables prior to and after the impact of the event by observing the average treatment effects of the treated (ATT). Moreover, the coefficient of interest, symbolized as $\beta_3$, presents the mean treatment effect generated by the event impact. In this paper, the proposed coefficient indicates the effect of COVID-19's public health emergency on the dissemination strength of pandemic-relevant video content, which is expected to be positive as the number of people using social media to collect information increases in face of public health emergency.

The dependent/explained variables in this study consist of video views, likes, and comments. Primarily, video views can, to a certain degree, indicate the extent of video dissemination. The number of likes on a video represents the audience's level of appreciation for the published content. Additionally, the number of comments on a video reflects the enthusiasm

of the audience to engage in topic discussions, which in turn captures the level of attention garnered by the video content [49,50]. Consistent with this, a holistic evaluation of the individual videos' news dissemination strength is achieved by comprehensively considering these three variables. The proposed comprehensive approach enables the researchers to dynamically observe shifts in the related videos' news dissemination strength during public health emergencies, aiming to identify the underlying trends.

Table 1 presents the summary statistics of this study. The research data for video views, comments, and likes are as of 30 June 2022. Given the disparity between the study period and the time of data collection, the collected data may not accurately reflect the situation at the time of the events. Therefore, the authors undertook a continuous 10-week tracking observation of videos posted by video bloggers in the sample within one day, totaling 1678 YouTube videos, in order to acquire an explicit understanding of the significance represented by the data. The results depict that the proportions of views, likes, and comments during the first week accounted for 85.83%, 95.35%, and 99.17% respectively of the total over the 10 weeks. This reveals that most of the video views, comments, and likes happen within the first week of video publication. Therefore, although accurate information regarding the certain time frame of video views, likes, and comments cannot be obtained from the sample, there is a reason to believe that a considerable portion of the views, likes, and comments occur shortly after the publication of video content, hence offering an approximate reflection of the video's dissemination during that period.

**Table 1. Summary statistics.**

| Variable | Mean | Std. Dev. | Minimum | Maximum | Observations[b] |
|---|---|---|---|---|---|
| Views | 87152.07 | 705066.63 | 3.00 | 51597032.00 | 17018 |
| ln(views) | 9.35 | 2.17 | 1.10 | 17.76 | 17018 |
| Likes | 1115.05 | 9192.93 | 0.00 | 807377.00 | 16395 |
| ln(likes) | 4.97 | 2.15 | 0.00 | 13.60 | 16138 |
| Comments | 190.87 | 698.20 | 0.00 | 37394.00 | 15743 |
| ln(comments) | 3.72 | 1.93 | 0.00 | 10.53 | 13871 |
| Videos related to COVID-19 in mainland China (%) | 41.31 | | | | 17018 |
| Videos from news & political bloggers (%) | 77.67 | | | | 17018 |
| Videos from personal media (%) | 46.76 | | | | 17018 |
| Videos from local media (%) | 36.47 | | | | 17018 |
| Views | Videos related to COVID-19 in mainland China (Treatment group) | | Videos related to mainland China (Control groups) | | Mean Difference |
| | Mean | Observations | Mean | Observations | |
| Before Natianal PHE | 29459.96 | 314 | 98086.21 | 4714 | -68626.25 |
| After Natianal PHE | 63879.04 | 6716 | 110450.03 | 5274 | -46570.98***[a] |
| Likes | Treatment group | | Control groups | | Mean Difference |
| | Mean | Observations | Mean | Observations | |
| Before Natianal PHE | 445.86 | 281 | 1340.84 | 4556 | -894.97 |
| After Natianal PHE | 754.45 | 6454 | 1406.35 | 5104 | -651.90*** |
| Comments | Treatment group | | Control groups | | Mean Difference |
| | Mean | Observations | Mean | Observations | |
| Before Natianal PHE | 85.35 | 303 | 173.40 | 4402 | -88.05** |
| After Natianal PHE | 229.40 | 6386 | 161.37 | 4652 | 68.03*** |

[a] **, *** denotes significance levels at the 5%, and 1%, respectively.

[b] The time range of the sample is from December 31, 2019, to February 20, 2020. Unless otherwise specified, the same applies to the following tables and figures.

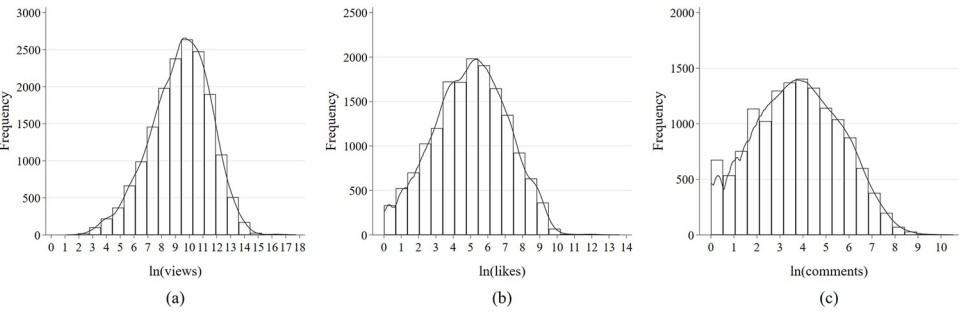

**Fig 1. Frequency distribution of the dependent variables.**

The upper section of Table 1 indicates that the average video views in the sample stand at 83,500, accompanied by a mean of 1,078.05 likes and 187.86 comments. Notably, nearly 41.92% of the sample videos are linked to the topic of the COVID-19 pandemic in mainland China. In addition to this, 77.39% of the video content is derived from the "News & Politics" category, 46.67% are from personal media channels, and 36.75% are from mainland China's media sources. Furthermore, the lower section of Table 1 shows the video views, comments, and likes, as well as the mean differences between the treatment and control groups for different periods of time. In the subsequent empirical analysis, the DiD method is used to determine the effect of event shocks. On the one end, the treatment group consists of news reports linked to the COVID-19 pandemic in mainland China; while on the other end, the control group includes other videos associated with topics in mainland China. Prior to the COVID-19 national PHE (termed as the first phase in the Chinese White Paper), the treatment group constituted 6.25% of the total sample for that time period. Conversely, the proportion of the treatment group increased to 54.01% during the COVID-19 PHE (referred to as the second stage in the Chinese White Paper). This indicates that public attention to the COVID-19 outbreak rapidly increased during the PHE, followed by an increase in video bloggers sharing related news content. Besides this, Fig 1 outlines the distribution of log-transformed video views, likes, and comments. Noticeably, all three major outcome variables demonstrate a normal distribution.

## Research results

This paper first presents the benchmark regression results, followed by parallel trends test and event study. Afterward, a robustness analysis is carried out, and finally, the heterogeneous effects of different types of bloggers are analyzed in this study.

### The impact of PHE on relevant new dissemination strength

Table 2 represents the influence of COVID-19 PHE on the dissemination strength of relevant videos. The sample period ranges from 31st Dec 2019 to 20th Feb 2020. The dependent variables are the logarithms of video views, likes, and comments, respectively. In addition, control variables contain information such as the number of subscribers for the video bloggers, language (Simplified Chinese/Traditional Chinese), video blogger categories, location, and dummy variables for video bloggers, video themes, and video posting times. Further, the standard errors (S.Es) are adjusted through video blogger-level clustering. In subsequent regression analyses, the control variables and clustered SEs remain consistent. As a consequence, the benchmark regression results align with the study's expectations. During the COVID-19 National PHE, the video views concerning the COVID-19 pandemic increased by 50.77% points, and video likes surged by 48.92% points. Parallel to this, video comments increased by

**Table 2. The impact of public health emergency on relevant new dissemination strength.**

| Variables | ln(views) | ln(likes) | ln(comments) |
|---|---|---|---|
| Videos related to COVID-19 in mainland China * public health emergency (PHE) | 0.5421***[a] | 0.5080*** | 0.4328*** |
| | (0.0991)[b] | (0.0989) | (0.1219) |
| Control variables | Yes | Yes | Yes |
| Fixed effect | Yes | Yes | Yes |
| Observations | 17018 | 16138 | 13871 |
| R2 | 0.701 | 0.731 | 0.624 |

[a]*, **, *** denotes significance levels at the 10%, 5%, and 1%, respectively.

[b]The regression model incorporates video blogger level clustered robust standard errors. Unless otherwise specified, the same applies to the following tables and figures.

40.32% points. The discussed effects are significantly different from zero (0) at the 1% level of statistical significance.

## Parallel trends test and event study

In this research, the DiD model is adopted to estimate the influence of public health emergency (PHE) on the dissemination strength of related videos. Reportedly, the baseline regression estimations depict that the relevant video's dissemination strength surged during the public health emergency. Subsequently, an event study methodology is adopted to test the parallel trends assumption and observe the dynamic effects of PHE, in order to ensure the reliability of the identification results. The specific model is expressed in Eq (2) as follows:

$$Y_{ijtl} = \beta_0 + \beta_k \cdot \sum_{k \geq -3}^{12} D_{t_{i0}+k} + \beta_2 \cdot X'_{ijdl} + u_j + u_d + u_l + \varepsilon_{ijdl} \tag{2}$$

Where $D_{t_{i0}+k}$ stands for a group of dummy variables, with the subscript $t_{i0}+k$ signifying the week in which a COVID-19-related video i was published on a YouTube channel. Correspondingly, $D_{t_{i0}+k}$ equates to 1 in case $t—t_{i0} = k$, otherwise, it equals 0. Moreover, 20th Jan is taken as the beginning of the public health emergency, as the first day of week 0; the week prior to PHE (k = -1) is the omitted reference group in Eq (2). Additionally, the group of parameters $\beta_k$ in the model reflects the impacts of PHE on the dissemination strength of specific news, relative to the reference week.

Fig 2 illustrates the results of the parallel trends test. Evidently, the coefficients of $\beta_k$ are not significantly different from 0 in the 2nd and 3rd weeks prior to the PHE. This implies that there are no significant variances in the strength of news dissemination between the treatment

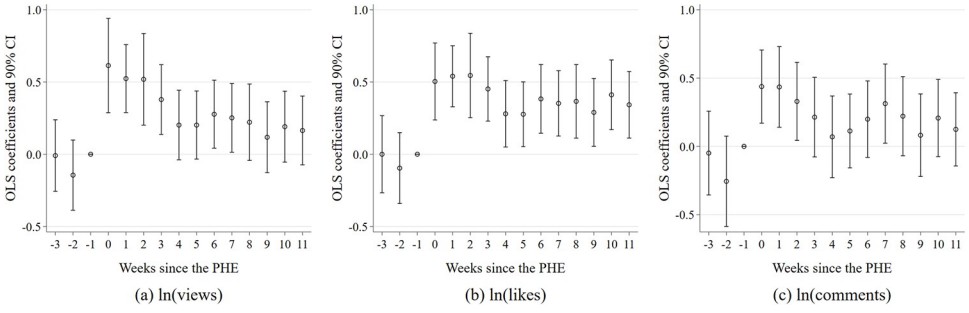

**Fig 2. Parallel trend test and event study.**

and control groups prior to the public health emergency, consequently confirming the assumption of parallel trends.

However, there was substantial increase in the views, likes, and comments of relevant videos during the COVID-19 National PHE, with substantial variance in this impact over time. Notably, the coefficients of $\beta_k$ were significantly positive for regression estimations with video views, likes, and comments as explained variables in the first three weeks of the COVID-19 National PHE. In particular, the aforementioned effect on the dissemination strength of relevant videos was strongest in the first week of the COVID-19 National PHE, with a 62.33% point increment in video views and a 45.08% point increase in comments. The proposed influences were significantly different from zero (0) at the 1% significance level. Successively, there was a gradual decline in the increase of dissemination strength brought about by the COVID-19 National PHE. Beginning from the 4th week, the potential effect of COVID-19 National PHE on video comments was no longer consistently significant. This suggests that the attention of audiences to the event was no longer significantly heightened as their enthusiasm to participate in the topic discussions was diminished over time. Subsequently, starting from the fifth week, the influence of COVID-19 National PHE on relevant video views ceased to be significant, which confirms that the significant expansion effect on the video's outreach was no longer existent.

Furthermore, it is pertinent to mention that the number of likes on relevant videos increased by 51.38 percentage points during the initial week of the COVID-19 National Public Health Emergency (PHE). This growth continued to expand in the subsequent 2nd and 3rd weeks. Afterward, though there were fluctuations, the number of video likes remained significantly higher throughout the entire sample period compared to the level prior to the COVID-19 National PHE.

## Robustness test

Firstly, various samples are created based on different thresholds of subscription count. Previously, among all Chinese-speaking video bloggers closely linked to mainland China, a sample with a subscription count of 10,000 or more was chosen by the authors. At present, the lower limit for sample subscription counts has been adjusted to 1,000 or 50,000, and the corresponding regression results are presented in Table 3(A) and 3(B), respectively. Furthermore, the sampling timeframe remains consistent for both scenarios, spanning from 31st Dec 2019 to 20th Feb 2020. It is evident that all interaction term coefficients are statistically significant, whereas the magnitudes and directions of the coefficients are the same as those in Table 2.

Secondly, the researchers consider adopting different criteria to identify video bloggers with a stronger association with mainland China. In the previous approach, bloggers who had 10% or more of their videos related to mainland China topics were selected. Presently, the outcomes are generated when this relevance threshold is adjusted to 5% and 15%, respectively. The regression outcomes are reported in Table 3(C) and 3(D). Correspondingly, the significance levels and magnitude of the interaction term coefficients in these cases align closely with the coefficients in Table 2.

Thirdly, the sample period is extended to cover a time period from 1st Sept 2019 to 28th April 2020. In Table 3(E), the sample includes videos published between 1st Sept 2019 and 20th Feb 2020. Owing to the absence of COVID-19 pandemic news reports prior to 31st 2019, videos associated with similar contagious diseases are taken as the treatment group. Moreover, Table 3(F) further expands the sample period to cover a time period from 31st Dec 2019 to 28th April 2020, thus encompassing the 1st to 4th phases of the COVID-19 pandemic. Prominently, the regression results for these two sets of samples are comparatively similar to those in

**Table 3. Robustness test.**

| Variables | ln(views) | ln(likes) | ln(comments) |
|---|---|---|---|
| Samples with subscription counts greater than or equal to 1000 | | | |
| Videos related to COVID-19 in mainland China * public health emergency (PHE) | 0.4831*** | 0.4879*** | 0.4252*** |
| | (0.1133) | (0.0983) | (0.1214) |
| Observations | 17675 | 16610 | 14141 |
| Samples with subscription counts greater than or equal to 50,000 | | | |
| Videos related to COVID-19 in mainland China * public health emergency (PHE) | 0.6035*** | 0.5558*** | 0.4429*** |
| | (0.1053) | (0.1088) | (0.1322) |
| Observations | 14566 | 13854 | 11984 |
| Bloggers whose videos contained 5% or more of content related to mainland China | | | |
| Videos related to COVID-19 in mainland China * public health emergency (PHE) | 0.5488*** | 0.5190*** | 0.4364*** |
| | (0.0990) | (0.0989) | (0.1219) |
| Observations | 17095 | 16215 | 13920 |
| Bloggers whose videos contained 15% or more of content related to mainland China | | | |
| Videos related to COVID-19 in mainland China * public health emergency (PHE) | 0.5937*** | 0.5674*** | 0.4828*** |
| | (0.1012) | (0.1013) | (0.1251) |
| Observations | 15853 | 15040 | 12889 |
| Samples from September 1, 2019, to February 20, 2020 | | | |
| Videos related to COVID-19 in mainland China * public health emergency (PHE) | 0.5060*** | 0.4083*** | 0.4020*** |
| | (0.1266) | (0.1273) | (0.1148) |
| Observations | 47090 | 44549 | 39584 |
| Samples from December 31, 2019, to April 28, 2020 | | | |
| Videos related to COVID-19 in mainland China * public health emergency (PHE) | 0.3366*** | 0.3695*** | 0.2995** |
| | (0.1064) | (0.1046) | (0.1248) |
| Observations | 46564 | 44357 | 38936 |
| All video samples from December 31, 2019, to February 20, 2020 | | | |
| Videos related to COVID-19 in mainland China * public health emergency (PHE) | 0.6106*** | 0.6450*** | 0.5734*** |
| | (0.1039) | (0.1117) | (0.1388) |
| Observations | 31396 | 29638 | 25727 |

Table 2. Additionally, the second set of samples, which contains a longer period of COVID-19 National PHE, exhibits a slightly diminished comprehensive influence due to the extended time frame.

Fourthly, the scope of the sample is expanded to include video content unrelated to mainland China. Initially, this study chose video content associated with mainland China as the analytical sample for the baseline regression, in order to satisfy the parallel trend assumption. Currently, a new regression is conducted without such a restriction. Consistently, Table 3(G) consists of all video samples from 31st Dec 2019 to 20th Feb 2020. Noticeably, all interaction term coefficients are statistically significant while the potential effect on videos related to PHE is similar to the baseline regression outcomes.

Finally, a placebo test is conducted to mitigate possible concerns that the significance of 3 statistical indicators emphasized in this paper was driven by certain random factors. Particularly, a "treatment group" and a "control group" are simulated in the original sample. Firstly, the authors randomly select a group of video bloggers, assuming that these bloggers had published videos related to the COVID-19 pandemic during the public health emergency (PHE). Next, simulated "treatment" groups and "control" groups are randomly chosen in stages, within the videos published by these bloggers. In fact, the number of simulated video bloggers who have published videos on the COVID-19 pandemic during each phase matches the actual

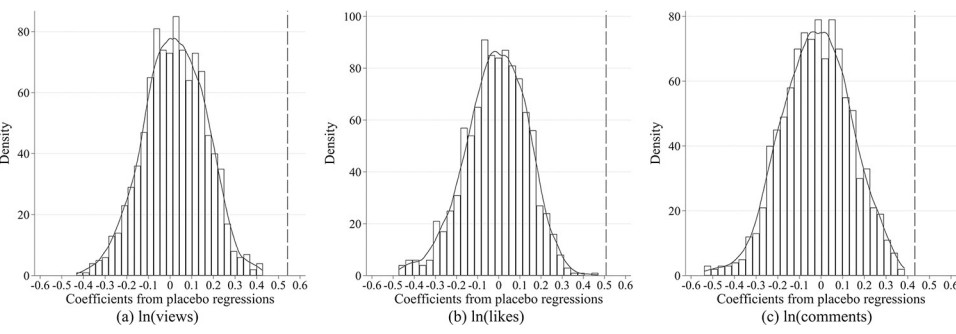

**Fig 3. Placebo test.**

sample data. In addition to this, the sample size of the simulated control and treatment groups in each stage matches the actual case. Afterward, a regression analysis is performed with the simulated sample. The proposed process is repeated 1000 times to obtain 1000 regression coefficients, while their distribution is illustrated in Fig 3. Specifically, the vertical line represents the magnitude of the corresponding coefficient in Table 2. The results reveal that the significance of the benchmark regression coefficients is robust from the statistical perspective.

## Heterogeneity analysis

The COVID-19 National PHE exerts a positive effect on the dissemination strength of relevant video content. However, this influence is likely to differ for videos posted by various categories of bloggers. Therefore, video bloggers are classified based on their characteristics into subgroups such as local media versus foreign media, news & political bloggers versus lifestyle bloggers, and personal media versus institutional media. Furthermore, each of these subgroups is separately expressed using Eq (1), in order to report the potential heterogeneities among them. In fact, this analysis attempts to unveil the characteristics of variations in dissemination strength for different types of bloggers during the COVID-19 National PHE.

Firstly, the samples are divided into two groups based on the primary content of the bloggers' videos: news & political bloggers and lifestyle bloggers. Thereafter, separate estimations are undertaken using Eq (1). In Fig 4, the first row displays samples from news & political bloggers, while the second row represents samples from lifestyle bloggers. Simultaneously, the numbers in the figure present the coefficients of interest, which are denoted as $\beta_3$ in Eq (1). Accordingly, for a certain sub-group, these coefficients indicate the effect of COVID-19 PHE on the dissemination strength of pandemic-related videos. Evidently, regardless of estimation in terms of views, likes, or comments, the dissemination strength of related video content

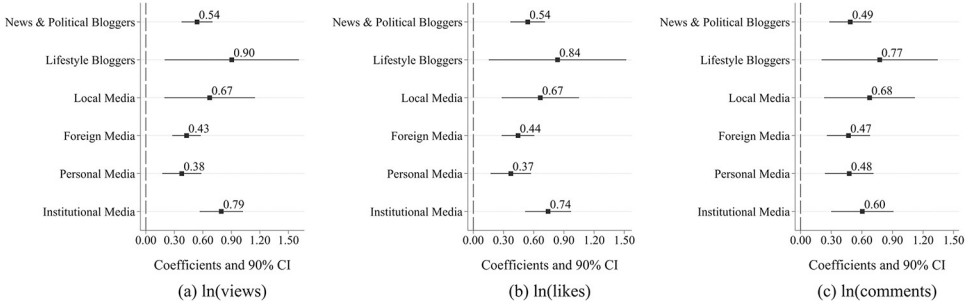

**Fig 4. Regression result of heterogeneity analysis.**

posted by news & political bloggers exhibited a lower growth during the COVID-19 National PHE, as compared to the lifestyle bloggers. Specifically, for the former, the growth in views of relevant videos amplified by 53.85% points, likes rose by 54.08% points, and comments augmented by 48.70% points during the COVID-19 National PHE. Contrarily, for the latter, the growth in views of related videos uplifted by 90.44% points, likes increased by 83.68% points, and comments increased by 77.44% points during the COVID-19 National PHE. The proposed effects were all significantly different from zero (0) at the 5% or 1% level of statistical significance.

Secondly, the samples are categorized into two groups based on the bloggers' location: local media (mainland China's media) and foreign media. Subsequent to this, Eq (1) is estimated for each group separately. As depicted in Fig 4, the 3rd row displays samples from local media, whereas the 4th row displays samples from foreign media. By analyzing the figure, it is obvious that the local media experienced a significantly greater increase in news dissemination strength than the foreign media during the COVID-19 National PHE. In specific, the views of relevant videos published by local media increased by 67.23% points. Likewise, likes increased by 66.58 percentage points, while comments increased by 67.57 percentage points. The aforementioned magnitudes are much higher than those of foreign media. These influences are all significantly different from zero (0) at the 1 or 5 percent level of significance.

Finally, given the variations in audience preferences and content styles between personal media and institutional media, this paper hypothesizes that personal- and institutional media may demonstrate varying amount of growth in news dissemination strength during the COVID-19 National Public Health Emergency (PHE). In Fig 4, the 5th row forecasts the results for personal media, whereas the 6th row depicts the results for institutional media. Obviously, institutional media exhibited a significantly higher increase in the distribution of relevant video content than personal media during the period of a public health emergency. Particularly, the relevant video views and likes published by institutional media increased by 79.44% and 74.35% points during the Covid-19 PHE, respectively, while the corresponding figures for personal media stood at 37.84 and 37.26 percentage points. In addition, these impacts are all significantly different from zero at the 1% level of statistical significance.

## Discussion

In the context of big data from YouTube, this research conducts a detailed observation of news dissemination within the overseas Chinese-speaking community during the COVID-19 National PHE. The study findings reflect that the views, likes, and comments of relevant videos during COVID-19 demonstrated a significant level of increase. In line with this, the public's response to the PHE has been highly rapid: within the first week of the PHE, views and comments of pandemic-associated videos increased by 62.33 and 45.08% points, respectively, thereby reaching their peak growth, followed by a gradual decline over time. Similarly, the increase in video likes reached 51.38% points in the first week, with the increment expanding in the subsequent two weeks. However, starting from the 4th and 5th weeks of the COVID-19 National PHE, there was no longer a significant growth in video comments and views associated with the PHE. Nevertheless, video likes remained significantly different from zero (0) throughout the period of a public health emergency.

Generally, the aforementioned regression results are consistent with study expectations. The public's demand for information regarding the PHE surged during the initial phase of the COVID-19 National Public Health Emergency (PHE), thereby leading to a rapid increase in the dissemination of relevant video content [51]. Nevertheless, the initially heightened dissemination strength of relevant videos was short-lived, returning to normal levels within 4 weeks.

Combining the observed phenomenon with extant literature, this study speculates that the lack of PHE's sustained coverage in news reports principally stems from the rapid shift in public attention instead of the media's subjective behavior [16,38]. Since audience focus shifts from the PHE over time, therefore news media, specifically online media adjust their content to match audience interests and stop posting PHE-related videos. As a result, there is a rapid decline in the coverage of the PHE shortly after its initial high-interest stage, in spite of the persistent existence of public risks.

Simultaneously, the influence of PHE led to a sustained and significantly positive uplift in the number of likes for related videos. This suggests that when reporting on public health emergencies (PHE), YouTube video bloggers attract an increasing number of viewers who identify with or are entertained by their video content. In addition to this, the audience attracted at this time demonstrated a certain level of stickiness: once engaged with a video blogger's content due to their coverage of the COVID-19 National PHE, audiences continue to support and follow the blogger due to their satisfaction with the content and alignment with the blogger's viewpoints. Further, some of the audience may subscribe to these channels, which confirms the incentive mechanisms of social media platforms to promote relevant news dissemination during PHEs.

Furthermore, the study findings established the heterogeneity in news dissemination of different categories of bloggers during the PHE. Particularly, lifestyle bloggers, local media, and institutional media display a significantly greater increase in dissemination strength during the COVID-19 National PHE, as compared to the news & political bloggers, foreign media, and personal media, respectively. Firstly, there exist varying degrees of growth in the transmission of relevant videos posted by lifestyle bloggers and news & political bloggers during the COVID-19 National PHE. The major reason for this lies in the differences in their regular content. Apparently, the COVID-19 PHE is the topic of interest in this research study. The coverage and comments on this event are considered as the regular content of news & political bloggers, while for other bloggers, this represents a way of tracking trending topics. The videos of the former shall spread more broadly within their prevailing audience base, whereas the videos of the latter shall reach several new viewers who otherwise are not in their existing audience base. Thus, this leads to a notable uplift in metrics such as views, likes, and comments for relevant videos uploaded by lifestyle bloggers during the COVID-19 PHE. Moreover, another possible explanation for this phenomenon is as follows: Given the greater focus on the COVID-19 National PHE, YouTube viewers are actively searching for a variety of information resources to obtain a deeper understanding of the event. Contrary to the profoundly homogenized professional news coverage, viewers might show a willingness to watch lifestyle bloggers who offer differentiated coverage and commentary on the event from unconventional and novel perspectives. As a result, the potential reach garnered attention, and the identification of the latter's relevant videos experienced a relatively higher surge during the COVID-19 National PHE.

Secondly, a substantial difference emerged in the growth of news dissemination strength between local and foreign media during the COVID-19 National PHE which is largely attributed to the localized benefits of mainland China's media. As the initial outbreak of the COVID-19 pandemic took place in Wuhan, mainland China's media were granted the capability to access firsthand information and instantly convey such information to overseas audiences, thus establishing a pronounced edge in terms of information timeliness. As a result, relevant videos from local media demonstrated an additional capacity to amass heightened attention, attract a larger viewership, and receive more prominent recognition during the COVID-19 National PHE.

Finally, during the pandemic PHE, variations in the surge of dissemination strength between institutional and personal media arose due to their distinctive traits. Previous studies have established that YouTube serves as both a source of useful and misleading information during PHE [52]. Deceptive information significantly originates from personal media, whereas the emergence of misleading information from official media is relatively scarce [53–55]. The research findings put forward in this study are consistent with the existing literature: In the process of gathering information on significant matters such as the COVID-19 National PHE that cost humans' health and even their lives, audiences significantly emphasize the reliability and credibility of information sources. Therefore, the audiences are more inclined to obtain information from institutional media; thereby, leading to a larger enhancement in the scope of video dissemination and audience involvement for such type of content.

## Conclusion and policy implications

This research paper reports a significant rise in the dissemination strength of relevant video content during the National PHE of COVID-19. Both the video views and comments for relevant content reached their peak growth within the first week of the PHE, thereafter gradually diminishing, and returning to pre-PHE levels within a period of one month. Additionally, the influence of the PHE differed among various types of video bloggers. Notably, during the COVID-19 PHE, lifestyle bloggers, local media, and institutional media displayed a significantly greater increase in the dissemination strength, as compared to the news & political bloggers, foreign media, and personal media, respectively.

These research findings put forward valuable policy implications. Firstly, the public pays due attention to the developments of the event during the National PHE of COVID-19, thereby leading to a significant rise in the dissemination of related news content. This increase sharply reaches its peak within the first week and gradually diminishes over time. This shows that the timeliness in information dissemination is crucial when health organizations respond to pandemics. Releasing accurate and explicit information promptly not only broadens the scope of information dissemination and realizes improved communication outcomes but also assists in quickly curtailing the spread of false information. Thus, health organizations must prioritize the timeliness of information release while disseminating information related to pandemics.

Secondly, the study findings reflect that during public health emergencies, the public relies more on information from institutional media, lifestyle bloggers, and local media, as compared to other media entities. The proposed categories of video bloggers play highly significant roles in the news dissemination of PHE while exerting a stronger influence on the viewpoints and attitudes of the public. Therefore, there is a dire need to leverage the potential of these media categories to promptly disseminate information, in order to address public health emergencies. Concurrently, the greater news dissemination strength of these media categories also implies that such media categories contribute to a substantial spread of false information, thus making these media classes imperative entities to focus on when countering false information during PHEs.

Lastly, the audience attracted through reporting relevant news displays a certain level of stickiness during public health emergencies. Meanwhile, some of the audience become subscribers to the channel, which confirms the incentive mechanisms of social media platforms to promote relevant news dissemination during PHEs. Therefore, as online media, it is of vital significance to ensure proactive participation in the dissemination of pertinent news during such PHE. This shall not only serve to enhance the public's comprehension of the situation but also amplify their impact and broaden the viewer base of the channel. In addition, health

organizations should collaborate with social media platforms to strengthen incentive mechanisms for the dissemination of precise information and encourage deeper involvement of online media in the propagation of authentic information.

This article highlights crucial research findings that offer insights into how health organizations can effectively disseminate relevant information in upcoming public health emergencies. Nonetheless, this study possesses inherent limitations. For instance, our data utilization is confined. Specifically, the empirical analysis of this study relied exclusively on text, neglecting potential insights from videos and images. Videos and images, intuitively, may contain richer information, suggesting avenues for further research.

## Supporting information

**S1 Table. Examples of representative keywords.**
(DOCX)

**S1 Dataset. Microsoft Excel dataset file of YouTube videos.**
(XLSX)

## Author Contributions

**Conceptualization:** Guochang Zhao.

**Data curation:** Dan Sun.

**Formal analysis:** Dan Sun.

**Investigation:** Dan Sun.

**Methodology:** Guochang Zhao.

**Project administration:** Guochang Zhao.

**Software:** Dan Sun.

**Supervision:** Guochang Zhao.

**Writing – original draft:** Dan Sun.

**Writing – review & editing:** Guochang Zhao.

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
