## [Decision Letter · Decision Letter 0]

17 Oct 2023

PONE-D-23-29289News Dissemination in Video-Sharing Platform During Public Health Emergency (PHE): Evidence from Chinese-Speaking Vloggers on YouTubePLOS ONE

Dear Dr. Zhao,

Thank you for submitting your manuscript to PLOS ONE. After careful consideration, we feel that it has merit but does not fully meet PLOS ONE’s publication criteria as it currently stands. Therefore, we invite you to submit a revised version of the manuscript that addresses the points raised during the review process.

We look forward to receiving your revised manuscript.

Kind regards,

Wajdy Jum’ah Al-Awaida, Ph.D

Academic Editor

PLOS ONE

Journal Requirements:

2. In your Methods section, please include additional information about your dataset and ensure that you have included a statement specifying whether the collection and analysis method complied with the terms and conditions for the source of the data.

Reviewers' comments:

Reviewer's Responses to Questions

**Comments to the Author**

1. Is the manuscript technically sound, and do the data support the conclusions?

Reviewer #1: Yes

Reviewer #2: Yes

2. Has the statistical analysis been performed appropriately and rigorously? 

Reviewer #1: Yes

Reviewer #2: Yes

3. Have the authors made all data underlying the findings in their manuscript fully available?

Reviewer #1: Yes

Reviewer #2: Yes

4. Is the manuscript presented in an intelligible fashion and written in standard English?

Reviewer #1: No

Reviewer #2: Yes

5. Review Comments to the Author

Reviewer #1: I enjoyed reading you paper. It deals with an important topic, espicially in light of climate change and the likelihood of public health emergencies for which effective news dissemination. Attached please

my observations for you to consider.

Best,

Reviewer #2: Summary

Based on big data from YouTube, this research study takes the declaration of COVID-19 National Public Health Emergency (PHE) as the event impact and employs a DiD model to investigate the effect of PHE on the news dissemination strength of relevant videos. The paper is well written and the topic is relevant to public health

Introduction

Paragraph 5 line 5- authors briefly describes their study findings. I suggest removing information related to findings.

Introduction, Last paragraph line 3….typographical error *this should be sample size

Research Background and Methodology

2.1 The COVID-19 Outbreak-Background information on the COVID-19 Outbreak should be summarized to be as brief as possible and moved into the introduction section so this section can be reserved for the methodology of the research

2. Research Methodology

Paragraph 1 should be moved to the introduction section.

3 data source and processing

Paragraph 3 line 2. The authors state that The approach for building the sampling frame of YouTube video bloggers was established by repetitively searching for top keywords from top news in mainland China. Kindly include what these top keywords are in the manuscript.

Paragraph 3 line 12. The authors state that small samples were excluded due to limited influence and a minimum subscriber threshold was set. Kindly state the what this threshold limit is and a justification for choosing it if possible.

Paragraph 4 states that video bloggers whose published videos contain 10% or more content relevant topics of mainland China were selected. Kindly explain what constitutes “content relevant topics of mainland China”.

Summary statistics

Paragraph 1- The authors state that “the number of likes on a video represents the audience's level of liking or agreement with the published content, as well as their confidence in the content, source, and channel” This is subjective however, if this measure has been used in similar research the authors should clearly indicate this and provide adequate references.

6. PLOS authors have the option to publish the peer review history of their article (what does this mean?). If published, this will include your full peer review and any attached files.

Reviewer #1: No

Reviewer #2: No

---

## [Author Response · Author response to Decision Letter 0]

28 Oct 2023

Dear Reviewers,

We greatly appreciate your attentive review, valuable insights, and constructive recommendations, all of which has significantly improved the presentation of our manuscript. Taking your review comments into serious consideration, we have carefully revised our manuscript. 

Please find attached our responses to your comments. We earnestly appreciate your work and hope that our revisions live up to your expectation. Again, thank you very much for your comments and suggestions.

---

## [Editor Report · Decision Letter 1]

7 Nov 2023

The impact of Public Health Emergency (PHE) on the news dissemination strength: Evidence from Chinese-Speaking Vloggers on YouTube

PONE-D-23-29289R1

Dear Dr. Zhao,

We’re pleased to inform you that your manuscript has been judged scientifically suitable for publication and will be formally accepted for publication once it meets all outstanding technical requirements.

Kind regards,

Wajdy Jum’ah Al-Awaida, Ph.D

Academic Editor

PLOS ONE
---

## [Editor Report · Acceptance letter]

17 Nov 2023

PONE-D-23-29289R1 

The impact of Public Health Emergency (PHE) on the news dissemination strength: Evidence from Chinese-Speaking Vloggers on YouTube 

Dear Dr. Zhao:

I'm pleased to inform you that your manuscript has been deemed suitable for publication in PLOS ONE. Congratulations! Your manuscript is now with our production department. 

Kind regards, 

on behalf of

Prof. Wajdy Jum’ah Al-Awaida 

Academic Editor

PLOS ONE